# Characterization of Highly Irradiated ALPIDE Silicon Sensors

**Valentina Raskina** *,† **and Filip Křížek** *,†

Nuclear Physics Institute of the Czech Academy of Sciences, CZ-25068 Řež, Czech Republic

* Correspondence: raskival@fjfi.cvut.cz (V.R.); Filip.Krizek@cern.ch (F.K.)

† These authors contributed equally to this work.

**Abstract:** The ALICE (A Large Ion Collider Experiment) experiment at CERN will upgrade its Inner Tracking System (ITS) detector. The new ITS will consist of seven coaxial cylindrical layers of ALPIDE silicon sensors which are based on Monolithic Active Pixel Sensor (MAPS) technology. We have studied the radiation hardness of ALPIDE sensors using a 30 MeV proton beam provided by the cyclotron U-120M of the Nuclear Physics Institute of the Czech Academy of Sciences in Řež. In this paper, these long-term measurements will be described. After being irradiated up to the total ionization dose 2.7 Mrad and non-ionizing energy loss $2.7 \times 10^{13}$ 1 MeV $n_{eq} \cdot cm^{-2}$, ALPIDE sensors fulfill ITS upgrade project technical design requirements in terms of detection efficiency and fake-hit rate.

**Keywords:** ALICE ITS upgrade; ALPIDE; MAPS; silicon pixel; radiation hardness; cyclotron

## 1. Introduction

ALICE (A Large Ion Collider Experiment) [1] is a high-energy physics detector at the CERN Large Hadron Collider (LHC). It is designed to study strongly interacting matter in the regime of high-energy densities and temperatures which occur when ultra-relativistic heavy nuclei collide. Under these conditions quarks and gluons escape their confinement in hadrons and form Quark–Gluon Plasma (QGP) [2].

In 2019–2020, the LHC machine will be upgraded. This so-called second long shutdown will be followed by the Run 3 and Run 4 data-taking periods in which ALICE aims to perform detailed measurements of QGP properties using low transverse-momentum ($p_T$) open-heavy-flavor hadrons, quarkonia, light vector mesons, and low-mass di-leptons [3]. Since these channels have a very small signal-background ratio, large statistics with un-triggered running is needed. ALICE plans to read minimum bias events in a continuous readout mode. Better vertexing and tracking efficiency at low $p_T$ are needed, which requires significant upgrades of ALICE sub-detectors. Another motivation for the ALICE upgrade is the expected increase of delivered luminosity by a factor of 100 in Run 3 and Run 4.

## 2. Alice Inner Tracking System Upgrade

The ALICE upgrade program includes many sub-projects. This paper is related to the upgrade of the ALICE Inner Tracking System, ITS [4]. This detector is essential for tracking and vertex reconstruction. The main goals of the ITS upgrade are: (i) to improve impact-parameter resolution of reconstructed tracks (by a factor of 5 in the longitudinal direction and by a factor of 3 in the transverse direction), (ii) to improve tracking efficiency and $p_T$ resolution for charged tracks with $p_T$ less than $1 \, \text{GeV}/c$, (iii) to increase the readout rate, and (iv) to allow fast insertion and removal of the detector during the end of year technical stops. These goals will be achieved by shifting the first detector layer closer to the beam line from the current 39 mm to 23 mm, by reducing the pixel size, and by reducing

the material budget $X/X_0$ per layer from 1.14% to 0.3% for the three innermost layers and to about 1% for the outer layers. When compared to the current ITS [1], which has 6 cylindrical layers of silicon detectors based on 3 different technologies (pixel, drift, and strip), the new ITS will have 7 layers of silicon pixel sensors called ALPIDEs, see Figure 1. The three innermost layers form the Inner Barrel and the four outer layers form the Outer Barrel. The area covered by ALPIDEs will be about 10 m$^2$. In total, there will be about 24,000 sensors [5]. The upgraded ITS aims to read out data up to a rate of 100 kHz in Pb–Pb collisions and 1 MHz in pp collisions.

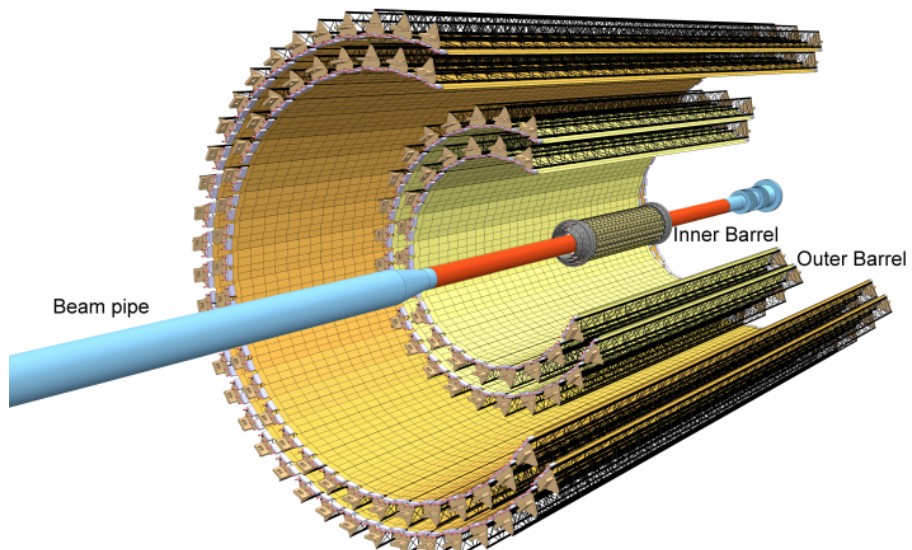

**Figure 1.** ALICE (A Large Ion Collider Experiment) Inner Tracking System (ITS) after upgrade, taken from [4].

## 3. The Alpide Sensor

The basic unit of the new ITS is ALPIDE [5–7], which stands for ALice PIxel DEtector. The sensor has a size of 1.5 × 3 cm. It is divided into 512 rows and 1024 columns of pixels with a pitch of 29.24 × 26.88 μm, which makes it possible to achieve a resolution better than 5 μm. The ALPIDE is a Monolithic Active Pixel Sensor (MAPS) which is based on the 180 nm CMOS technology of TowerJazz [8]. This technology uses up to 6 metal layers which in combination with small structure size enables to implement high density CMOS digital circuitry with low power consumption. The average power density across the sensor surface is less than 40 mW cm$^{-2}$ [5,7]. Another important feature is the implementation of a high-resistivity (>1 kΩ) epitaxial layer and a deep p-well, see Figure 2. The deep p-well layer prevents the collection of charge carriers by the n-well of pmos transistors, therefore both nmos and pmos transistors can be implemented in the active pixel area. The thickness of the epitaxial layer is 25 μm. The depletion volume can be increased by applying a moderate reverse bias voltage to the substrate. This also lowers the capacitance of the collection diode and results in a higher collection efficiency [7].

Each ALPIDE pixel contains a sensitive volume, as well as the front-end electronics. The electronics manage charge collection, signal amplification, and discrimination, and write binary hit information to an event buffer. There are several 8-bit DACs (Digital to Analog Converters) on the periphery of the chip, which regulate voltages and currents in the front-end circuits of pixels. The most relevant DACs for the work reported in this paper are voltage $V_{CASN}$ and current $I_{THR}$, which control charge threshold [6].

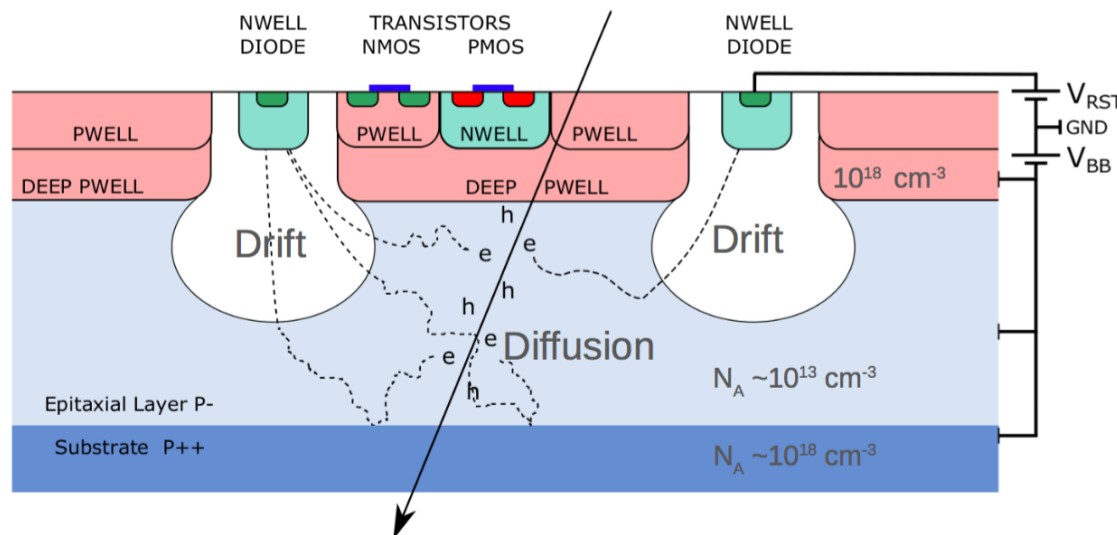

**Figure 2.** Cross section of an ALPIDE pixel. A charged particle crosses the sensitive volume (high-resistivity epitaxial layer between the substrate and the layer with CMOS transistors) and generates free charge carriers that diffuse across the epitaxial layer until they reach the drift region of a n-well diode, where they are being collected, taken from [4].

Table 1 shows the required parameters of ALPIDE from the technical design report [4] and the achieved performance. The table suggests that the performance of the sensor satisfies all requirements for the Inner and Outer Barrel. After Run 3 and Run 4, the expected total ionizing dose (TID) accumulated by a chip in the Inner Barrel will be 270 krad and the non-ionizing energy loss (NIEL) is expected to be $1.3 \times 10^{12}$ 1MeV $n_{eq} \cdot cm^{-2}$. However, we tested the ALPIDE chip under up to 10 times larger radiation loads. The corresponding study is reported in this paper.

**Table 1.** Requirements for a sensor in the Inner and Outer Inner Tracking System (ITS) Barrel and the achieved ALPIDE performance. Taken from [4].

|  | Inner Barrel | Outer Barrel | ALPIDE Performance |
|---|---|---|---|
| Thickness [µm] | 50 | 100 | OK |
| Spatial resolution [µm] | 5 | 10 | ~5 |
| Chip dimension [mm] | $15 \times 30$ | $15 \times 30$ | OK |
| Power density [mW/cm$^2$] | <300 | <100 | <40 |
| Event-time resolution [µs] | <30 | <30 | ~2 |
| Detection efficiency [%] | >99 | >99 | OK |
| Fake-hit rate [event$^{-1} \cdot$ pixel$^{-1}$] | <10$^{-6}$ | <10$^{-6}$ | <10$^{-10}$ |
| NIEL radiation tolerance [1 MeV $n_{eq} \cdot cm^{-2}$] | $1.7 \times 10^{13}$ | $3 \times 10^{10}$ | OK |
| TID radiation tolerance [krad] | 2700 | 100 | OK |

## 4. Radiation Hardness Tests at the Nuclear Physics Institute

Radiation hardness of ALPIDE sensors was tested using a 30 MeV proton beam provided by the U-120M cyclotron at the Nuclear Physics Institute of the Czech Academy of Sciences (NPI CAS) [9]. For this purpose we use an experimental setup which is shown in Figure 3. The extracted proton beam from the cyclotron has an energy of 34.8 MeV with an RMS of 0.3 MeV and passes through a beam line which is terminated with an energy degrader unit. This unit contains 5 aluminum plates with different thickness which can be inserted into the beam using a remotely controlled pneumatic system. The first aluminum plate is 8 mm thick and serves as a beam stop. Thickness of the second plate is 0.56 mm. This plate is used during the ALPIDE irradiation to make the beam profile wider such that

the ALPIDE sensor is irradiated more uniformly across its surface. The other 3 plates were not used in the experiment. The setup with the ALPIDE sensor is placed 130 cm from the end of the beam line. The setup is mounted on a remotely controllable, movable stage and consists of an ionization chamber, another aluminum beam stop plate which can shield the irradiated ALPIDE sensor, and a passive aluminum shielding that protects a readout board connected with the ALPIDE sensor.

The ionization chamber Farmer 30010 from PTW-Freiburg [10] is used to monitor the proton flux and is read out using a UNIDOS E Universal Dosemeter. The chamber has a sensitive volume of 0.6 cm$^3$ filled with air and provides a linear response to the incoming proton flux up to about $10^9$ proton· cm$^{-2}$· s$^{-1}$ [9]. The vertical distance between the center of the ionization chamber and the center of the ALPIDE sensor is measured using a laser tracker.

Based on GEANT4 simulation, it was estimated that the 0.56 mm thick energy degrader plate and the air decrease the beam energy from 35 MeV to 30 MeV before the beam hits the ALPIDE. The beam intensity profile is measured by moving the whole setup through the beam center along the horizontal and vertical direction. In both cases, the beam profile can be well described by a Gaussian with a standard deviation of about 22 mm. The dose absorbed in the irradiated sample is estimated based on the formula

$$\text{TID[krad]} = 1.602 \times 10^{-8} \times \text{S[MeV} \cdot \text{cm}^2 \cdot \text{mg}^{-1}] \times F[\text{cm}^{-2}], \tag{1}$$

where S is the linear energy transfer and *F* is the proton fluence [11]. The non-ionizing enegy loss induced by the 30 MeV proton beam is calculated as

$$\text{NIEL[1 MeV n}_{\text{eq}} \cdot \text{cm}^{-2}] = 2.346 \times F[\text{cm}^{-2}], \tag{2}$$

where the factor 2.346 is a tabled coefficient taken from [12].

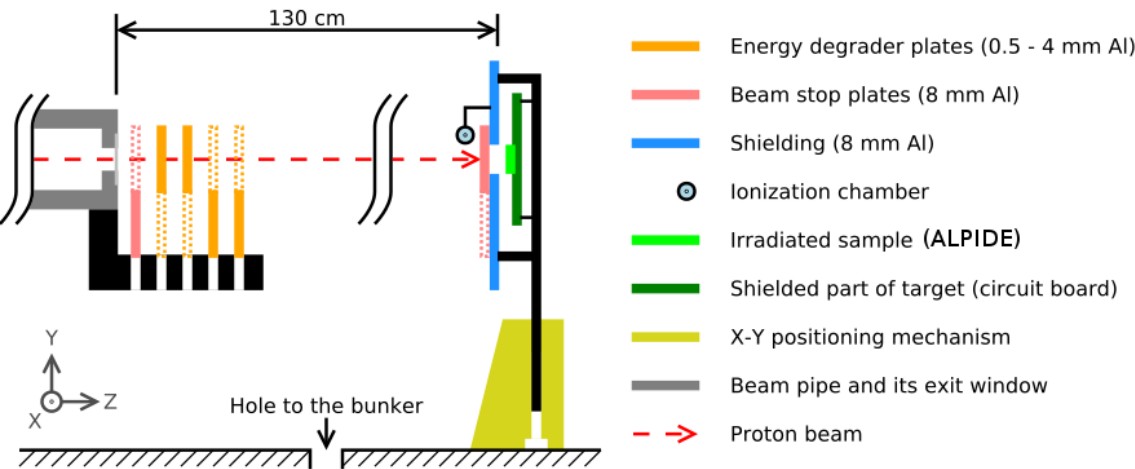

**Figure 3.** Sketch of the beam route from the beamline exit window to the irradiated sample through energy degrader plates, taken from [9].

Irradiation campaigns of each ALPIDE sensor took place every month since September 2016. ALPIDE was operated with $-3$ V reverse substrate bias. Sensors were irradiated with proton fluxes of the order of $10^8$ proton· cm$^{-2}$· s$^{-1}$. The ALPIDE irradiation was carried out as follows. The beam was interrupted periodically using the first beam stop plate. When the beam stop was out of the beam, the sensor was irradiated and analogue and digital currents consumed by the ALPIDE chip were monitored, see Figure 4. When the beam stop interrupted the beam, threshold and DAC scans were performed. The dependence of the absorbed total ionization dose and proton fluence on time for different irradiation campaigns for one chip is shown in Figure 5. The increasing trend in the curves corresponds to the irradiation and the flat trend corresponds to the period when the ALPIDE was not

irradiated. The first irradiation campaign that took place in September 2016 was the longest one and the accumulated dose was the highest. In the rest of the campaigns, the chip got about 100 krad, which corresponds to about one third of the total absorbed dose expected during Run 3 and Run 4. After each irradiation campaign, the chip was left to anneal at room temperature.

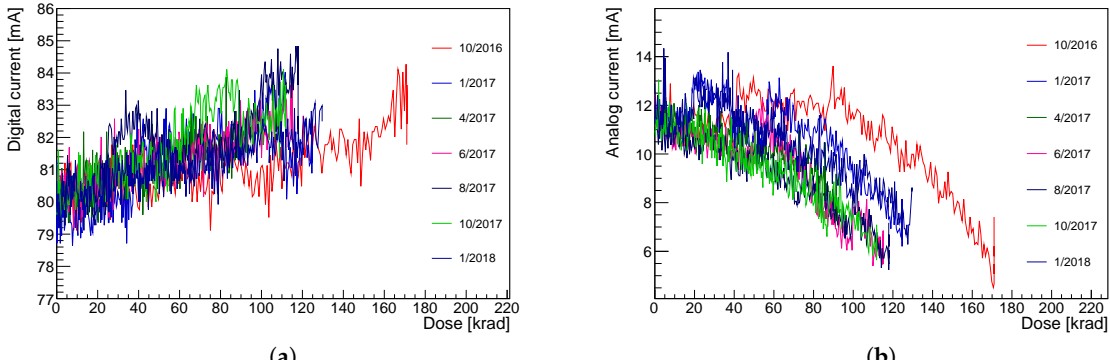

(**a**)　　　　　　　　　　　　　　　　　　　(**b**)

**Figure 4.** Digital (**a**) and analog (**b**) support currents versus total ionizing dose measured for different irradiation campaigns. The dates of the campaigns are listed in the legends.

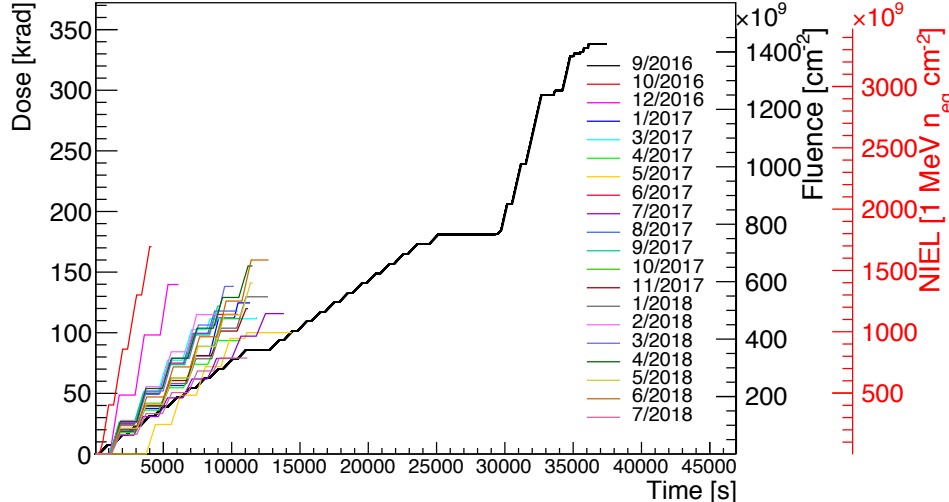

**Figure 5.** The total ionization dose, the accumulated proton fluence, and the non-ionizing energy loss (NIEL) for different irradiation campaigns. The legend gives the date of the performed campaigns.

In each pixel, charge threshold is measured by injection of a given charge from an injection capacitance to the pixel analog front end. The injection is repeated 50 times and the threshold is defined as a charge which is registered by a pixel with a 50% probability. In the case of ALPIDE sensors, the charge threshold depends mainly on two DACs: $I_{THR}$ which determines the shape of the pulse and $V_{CASN}$ which regulates the baseline voltage [6]. An example of the firing probability of a pixel versus the injected charge is shown in Figure 6. The dependence is fitted by the so-called S-curve

$$S(Q) = \frac{1}{2}\left(50 + 50 \times \mathrm{erf}\left(\frac{Q - Q_{THR}}{\sqrt{2}\sigma}\right)\right), \tag{3}$$

where $Q$ is injected charge and $Q_{THR}$ is threshold. The formula assumes that the temporal noise, which smears the threshold value, has a Gaussian character. The threshold distribution obtained from 10% of pixels for the default settings of the DACs for a non-irradiated chip is shown in Figure 7.

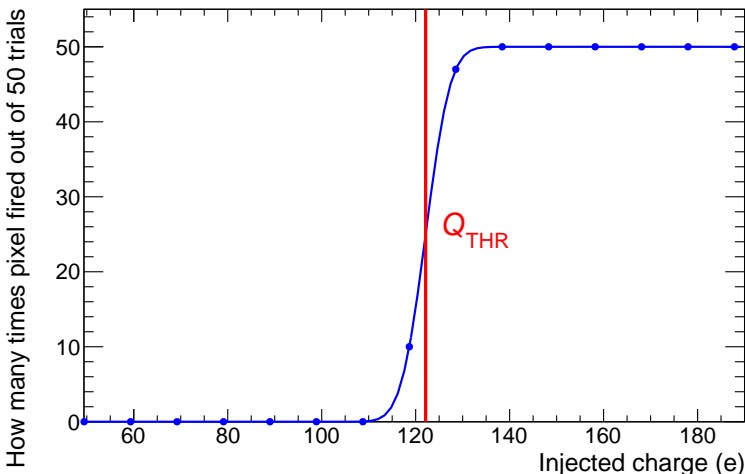

**Figure 6.** Probability of charge registration in an ALPIDE pixel. The data are fitted with the S-curve (3).

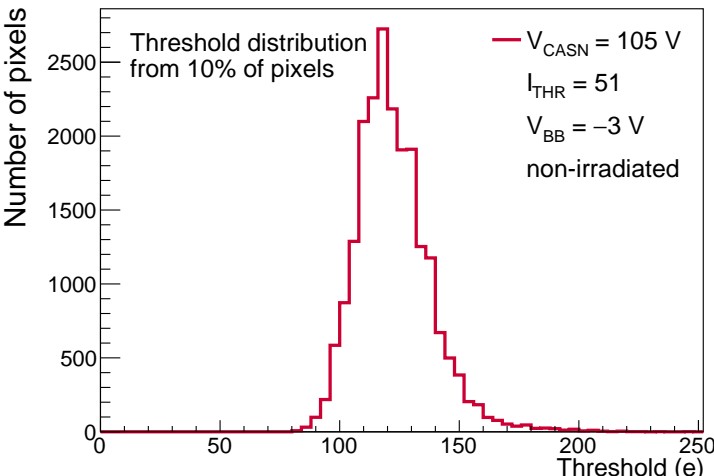

**Figure 7.** Distribution of charge thresholds from 10% of pixels of a non-irradiated chip.

Figure 8 shows the mean threshold as a function of the accumulated dose for different irradiation campaigns. Initially until October 2017, the ALPIDE was operated using the default DAC settings for $-3$ V reverse substrate back bias. In this period, the mean threshold was decreasing with the accumulated dose without any sign of annealing. In October 2017, the DAC settings of the chip were changed to increase the threshold and to suppress the noise. Since then we observe that the chip anneals after each campaign.

After obtaining the total ionizing dose of 2700 krad and the NIEL of $2.7 \times 10^{13}$ 1 MeV $n_{eq} \cdot cm^{-2}$, the chip was characterized at the CERN Proton Synchrotron. There the ALPIDE was tested using a 6 GeV/$c$ pion beam. The sensor was installed in a telescope which consisted of 7 planes of ALPIDE sensors. The tested ALPIDE (device under test, DUT) formed the middle plane of the telescope, the other ALPIDEs served as reference planes for pion track reconstruction. The EUTelescope software [13] was used to estimate the detection efficiency, which was obtained by comparing an extrapolated hit position in the DUT calculated from tracking planes with the actual measured hit position in the DUT.

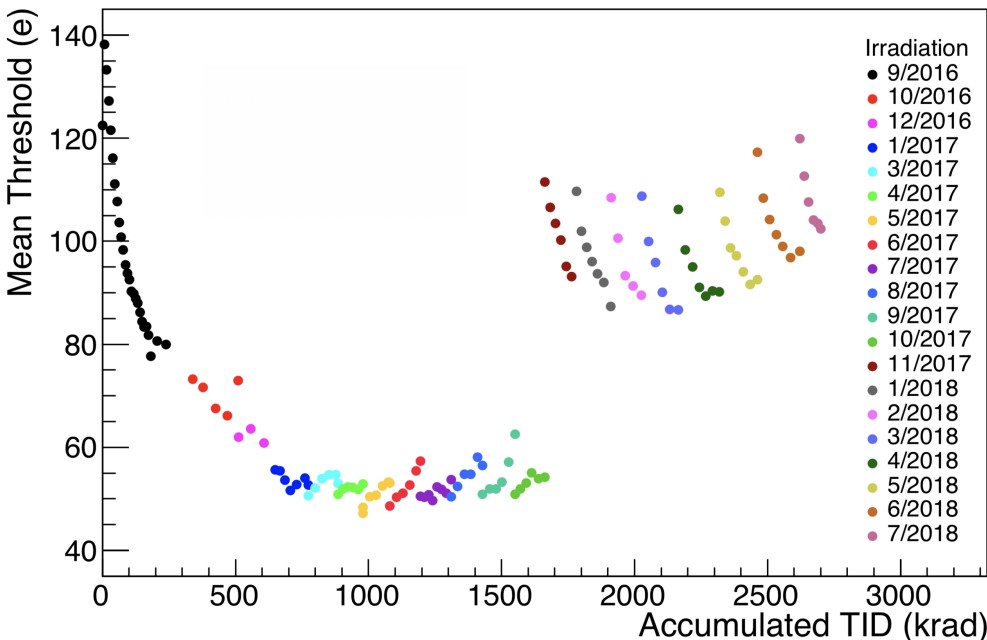

**Figure 8.** Mean threshold versus accumulated total ionizing dose. The dates of the performed campaigns are quoted in the legend.

Figure 9 shows the detection efficiency and the fake-hit rate as a function of the mean charge threshold for the irradiated chip and for a non-irradiated reference sensor. The red dash-dotted line corresponds to the project limit on fake-hit rate which is $10^{-6}$/pixel/event and the black dash-dotted line gives the limit on detection efficiency which should be higher than 99%. As it is seen from the figure, the irradiated sensor still fulfills the requirements of the upgrade project in terms of detection efficiency and the fake-hit rate in the threshold range $\approx$ 150–200 electrons.

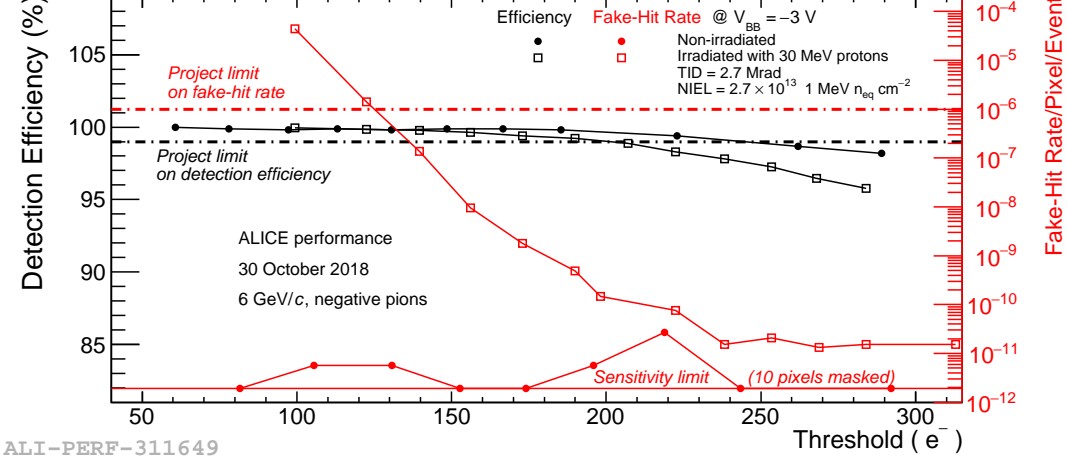

**Figure 9.** The detection efficiency and the fake-hit rate of the irradiated and non-irradiated ALPIDE sensors versus threshold charge for $-3$ V reverse substrate bias voltage. The detection efficiency was obtained using a 6 GeV/$c$ pion beam at CERN PS.

## 5. Conclusions

After reaching ten times the radiation load expected for an ALPIDE sensor in the Inner Barrel of ALICE ITS in Run 3 and Run 4, the sensor is still operational and fulfills project goal requirements. We observed that irradiation with 30 MeV protons caused a steady drop of the mean charge threshold

when the chip was operated with the nominal DAC settings for $-3\,\text{V}$ reverse substrated bias voltage. The initial threshold level could, however, be recovered by retuning DAC settings, which leads to suppression of noisy pixels.

**Author Contributions:** Conceptualization, F.K.; methodology, F.K.; software, V.R. and F.K.; validation, V.R. and F.K.; formal analysis, V.R. and F.K.; investigation, V.R. and F.K.; resources, F.K.; data curation, F.K.; writing—original draft preparation, V.R.; writing—review and editing, F.K.; visualization, V.R.; supervision, F.K.; project administration, F.K.; funding acquisition, F.K.

**Funding:** This research was funded by the Ministry of Education, Youth and Sports of the Czech Republic grant number LTT17018.

**Conflicts of Interest:** The authors declare no conflict of interest.

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
