# Peer review of "Characterization of Highly Irradiated ALPIDE Silicon Sensors"

_universe, doi:10.3390/universe5040091_

Reviewer 1 Report

This is nicely written paper reporting important measurements related to the performance ALPIDE sensors in high radiation environment . Text is clear and the results are well presented. I have few minor suggestions in order to improve the paper:

line 20: please add information why continuous readout is planned. what is the advantage to usual minimum bias triggered events. Why upgrades of detectors are needed for this.

line 54 does -> do; writes -> write

line 55 what DAC stands for?

line 68 please be more specific about 35 MeV, what is precision or range of energy of protons

line 83 typo estimated

I recommend the paper for publication.

Author Response

Point 1: line 20: please add information why continuous readout is planned. what is the advantage to usual minimum bias triggered events. Why upgrades of detectors are needed for this.

Response 1: Done

Point 2: line 54 does -> do; writes -> write

Response 2: Done

Point 3: line 55 what DAC stands for?

Response 3: DAC stands for Digital to Analog Converters

Point 4: line 68 please be more specific about 35 MeV, what is precision or range of energy of protons

Response 4: The proton beam energy is 34.8 MeV with the RMS of 0.3 MeV

Point 5: line 83 typo estimated

Response 5: Done

Reviewer 2 Report

The argument of the article is interesting and overall sufficiently clear. Nevertheless few points should be better clarified:

1) par. 4 lines 69-72: in Fig.3 there appear two Al plates, 0.56mm and 4mm thick, but in the text only the first one is mentioned: was the 4mm plate not used ? what would its effect be ?

2) par. 4 lines 84-86: how was the beam profile measured ? you say the setup was moved, but it is not clear how the beam intensity was actually measured (using the ionization chamber ? or the detector itself ?)

3) Fig.3 appears to be too much schematic and not so clear, it would be better to replace with a clearer one

4) it is not clear whether one single chip was irradiated many times, or many chips were tested. Did you change the chip between each campaign ?

5) par. 4 lines 96-97: you measured the analogue and digital currents; did you analyzed these data ? do you have results on current variation during irradiation ? adding these measures too would increase the significance of the article

6) Fig.6 : why the threshold of only the 10% of pixels is plotted ? how were these pixels chosen ? (I mean: did you take 10% of pixels randomly or did you chose them with some criterion, eg. same row/column ?)

7) Figs 5 & 6: from Fig.6 (and 7) it seems that a non-irradiated pixel has a threshold of ~130-140 e ; so in Fig.5 (comparing again with 7) the threshold of a highly-irradiated pixel is shown: if this is the case it should be clearly mentioned in the text and/or in the caption; alternatively the same curve of a non-irradiated pixel should be used

8) I would suggest some minor changes to the text:

i) line 2: concentric -> coaxial

ii) line 30: detection layer -> detector layer

iii) line 104: at the room temperature -> at room temperature

Author Response

1)par. 4 lines 69-72: in Fig.3 there appear two Al plates, 0.56mm and 4mm thick, but in the text only the first one is mentioned: was the 4mm plate not used ? what would its effect be ?

Response 1: In our experiment only 2 plates were used, first one to stop the beam and the second one to make the shape of the beam wider. The other 3 plates were not used in our experiment. I changed the text a little bit.

2) par. 4 lines 84-86: how was the beam profile measured ? you say the setup was moved, but it is not clear how the beam intensity was actually measured (using the ionization chamber ? or the detector itself ?)

Response 2: The beam was measured by ionisation chamber (lines 79-80), the setup with ionisation chamber,  ALPIDE and aluminium plate was moved along horizontal and vertical axises to measure the beam profile.

3) Fig.3 appears to be too much schematic and not so clear, it would be better to replace with a clearer one

Response 3: Fig. 3 shows the experimental setup and I really can not imagine, what can I cut from the picture and with which scheme it can be replaced. Do you think it is necessary?I can also add photo of this setup.

4) it is not clear whether one single chip was irradiated many times, or many chips were tested. Did you change the chip between each campaign ?

Response 4: Two chips were irradiated many times since September 2016 every month, in this work there are presented results for one chip, which has had already gained 2700 krad. I changed the text a little bit.

5) par. 4 lines 96-97: you measured the analogue and digital currents; did you analyzed these data ? do you have results on current variation during irradiation ? adding these measures too would increase the significance of the article

Response 5: Yes, I have analysed these data, I added the graph with behaviour of analog and digital supply current.

6) Fig.6 : why the threshold of only the 10% of pixels is plotted ? how were these pixels chosen ? (I mean: did you take 10% of pixels randomly or did you chose them with some criterion, eg. same row/column ?)

Response 6: When one runs threshold scan on a Alpide chip, typicialy it scanns just 10% of pixels. The pixels are chosen such that they are distributed over the matrix uniformly.

7) Figs 5 & 6: from Fig.6 (and 7) it seems that a non-irradiated pixel has a threshold of ~130-140 e ; so in Fig.5 (comparing again with 7) the threshold of a highly-irradiated pixel is shown: if this is the case it should be clearly mentioned in the text and/or in the caption; alternatively the same curve of a non-irradiated pixel should be used

Response 7: The curve was for irradiated chip, so I have changed the Fig. 5 , now it is for non-irradiated chip

8) I would suggest some minor changes to the text:

i) line 2: concentric -> coaxial

ii) line 30: detection layer -> detector layer

iii) line 104: at the room temperature -> at room temperature

Response 8: Done.